# Mechanistic and Structural Insights on Difluoromethyl-1,3,4-oxadiazole Inhibitors of HDAC6

**DOI:** 10.3390/ijms25115885

**Published:** 2024-05-28

**Authors:** Edoardo Cellupica, Aureliano Gaiassi, Ilaria Rocchio, Grazia Rovelli, Roberta Pomarico, Giovanni Sandrone, Gianluca Caprini, Paola Cordella, Cyprian Cukier, Gianluca Fossati, Mattia Marchini, Aleksandra Bebel, Cristina Airoldi, Alessandro Palmioli, Andrea Stevenazzi, Christian Steinkühler, Barbara Vergani

**Affiliations:** 1Research and Development, Italfarmaco Group, 20092 Milan, Italy; e.cellupica@italfarmacogroup.com (E.C.); a.gaiassi@italfarmacogroup.com (A.G.); i.rocchio@italfarmacogroup.com (I.R.); g.rovelli@italfarmacogroup.com (G.R.); r.pomarico@italfarmacogroup.com (R.P.); g.sandrone@italfarmacogroup.com (G.S.); g.caprini@italfarmacogroup.com (G.C.); p.cordella@italfarmacogroup.com (P.C.); g.fossati@italfarmacogroup.com (G.F.); m.marchini@italfarmacogroup.com (M.M.); a.stevenazzi@italfarmacogroup.com (A.S.); c.steinkuhler@italfarmacogroup.com (C.S.); 2Department of Biochemistry, Selvita S.A., 30-394 Kraków, Poland; cyprian.cukier@selvita.com (C.C.); aleksandra.bebel@selvita.com (A.B.); 3Department of Biotechnology and Biosciences, University of Milano-Bicocca, 20126 Milan, Italy; cristina.airoldi@unimib.it (C.A.); alessandro.palmioli@unimib.it (A.P.)

**Keywords:** histone deacetylase 6 (HDAC6), difluoromethyl-1,3,4-oxadiazole (DFMO), non hydroxamic inhibitors, NMR, LC-MS, X-ray crystallography, enzyme kinetics, DFMO hydrolysis, difluoroacetylhydrazide (DFAcH)

## Abstract

Histone deacetylase 6 (HDAC6) is increasingly recognized for its potential in targeted disease therapy. This study delves into the mechanistic and structural nuances of HDAC6 inhibition by difluoromethyl-1,3,4-oxadiazole (DFMO) derivatives, a class of non-hydroxamic inhibitors with remarkable selectivity and potency. Employing a combination of nuclear magnetic resonance (NMR) spectroscopy and liquid chromatography-mass spectrometry (LC-MS) kinetic experiments, comprehensive enzymatic characterizations, and X-ray crystallography, we dissect the intricate details of the DFMO-HDAC6 interaction dynamics. More specifically, we find that the chemical structure of a DMFO and the binding mode of its difluoroacetylhydrazide derivative are crucial in determining the predominant hydrolysis mechanism. Our findings provide additional insights into two different mechanisms of DFMO hydrolysis, thus contributing to a better understanding of the HDAC6 inhibition by oxadiazoles in disease modulation and therapeutic intervention.

## 1. Introduction

The mammalian histone deacetylase family comprises 18 different enzymes that can be grouped into two mechanistically distinct families: the 7 NAD-dependent members of the Sirtuin family and the 11 Zn-dependent Histone Deacetylases (HDACs). HDACs are classified into class I (HDAC1, 2, 3, and 8), class IIa (HDAC4, 5, 7, and 9), class IIb (HDAC6 and 10), and class IV (HDAC11), according to differences in structure, enzymatic function, subcellular localization, and expression pattern. HDACs catalyze the removal of acyl groups from lysine residues of histones, non-histone proteins, and polyamines [1,2,3,4], thus playing a crucial role in various diseases [5]. The dysregulation of HDAC activity, especially that of class I HDACs, has been related to the development of different kinds of cancers, both hematological and solid tumors. In fact, the alteration of their activity results in a more compact nucleosome structure, and, consequently, a downregulation of tumor suppressor genes [6]. Overexpression or mutation of HDACs has been described in breast cancer, colon cancer, medulloblastoma, and many others [7]. The alteration of HDACs activity is also involved in non-oncologic pathologies, such as neurodegenerative disorders [8], and autoimmune diseases [9]. 

The structure of HDAC6 stands out within the HDAC family due to the presence of two distinct functional catalytic domains (CD1 and CD2), both zinc-dependent and exhibiting differences in substrate preferences [10,11].

This hydrolase is predominantly located in the cytoplasm, where it deacetylates a variety of non-histone proteins, such as α-tubulin, cortactin, β-catenin, peroxiredoxin, heat shock protein 90 (HSP90), and others [12]. Being involved in regulating the activity of such a large number of non-histone substrates, the alteration in HDAC6 function can be responsible of the onset of many pathologies and diseases. The HDAC6 involvement in microtubule dynamics is particularly important, as its alteration has been related to axonal transport deficits, commonly observed in peripheral neuropathies, both genetically originated and chemotherapy-induced [13,14]. The role of HDAC6 in the process of catabolism of degraded proteins through the aggresome pathway is also crucial. Alteration of this cytoprotective activity has been described as a possible cause of the onset of neurodegenerative disorders such as Parkinson’s and Huntington’s disease [15]. Targeting HDAC6 has emerged as an attractive strategy in various pathologies, since its selective inhibition has shown superior tolerability and safety in preclinical models and human clinical trials compared to pan-HDAC inhibition [16]. However, the first generation of selective HDAC6 drug candidates was designed by using hydroxamic acid as the zinc binding group (ZBG), a potent warhead that, however, has metabolic stability and safety issues, limiting its therapeutic applications. 

A relatively large number of inhibitors in zebrafish HDAC6-CD2 (zHDAC6-CD2) crystal structures has been identified and investigated in the last decade, since the seminal works of Hai and Christianson [10] and Miyake et al. [17] were published. However, the coprecipitated inhibitors are hydroxamate derivatives, with the ZBG bound to an aromatic moiety (usually a six-membered ring), a pharmacophoric feature able to provide light HDAC6 selectivity per se. This property can be enhanced by fluorine decoration or by inclusion of a nitrogen atom [18] in the adjacent aromatic moiety. When the hydroxamic acid is bound to a phenyl ring, it usually exhibits a monodentate chelation mode, allowing room for the retention of a water molecule (catalytic water) in the metal coordination sphere, and already available in the apo protein. The monodentate coordination is usually provided by the anionic form of the hydroxamic OH group, whereas bidenticity is typically observed in complexes where the inhibitor exhibits an aliphatic chain [10,17,19] bound to the hydroxamate group and involves both oxygen atoms of the warhead, thus displacing the catalytic water from the Zn ion coordination sphere. 

Over the last few years, difluoromethyl-1,3,4-oxadiazole (DFMO) derivatives have emerged [20,21] as a new promising class of highly selective, non-hydroxamic HDAC6 inhibitor. Their exceptional selectivity [22] (greater than 10^4^-fold selectivity for HDAC6 compared to all other HDAC isozymes) encouraged several investigations into the inhibitory effects brought about by this novel class of molecules. These studies revealed that DFMOs act as mechanism-based, substrate-analog inhibitors of HDAC6 [22,23,24,25,26]. The inhibition mechanism involves the formation of a difluoroacetylhydrazide (DFAcH), which is generated in situ by the HDAC6-catalyzed hydrolysis of the DFMO warhead. This reaction involves the nucleophilic attack of the zinc-bound water to the sp^2^ carbon adjacent to the difluoromethyl moiety, followed by a ring opening of the oxadiazole. Subsequently, the DFAcH intermediate is further hydrolyzed to the corresponding final hydrazide. This secondary hydrolysis reaction is likely to be driven by the specific experimental conditions and is observed at relatively high enzyme and substrate concentrations in vitro.

Despite the increasing number of published studies and patents indicating the unique and almost absolute selectivity of DFMOs towards HDAC6, their binding mode to the enzyme remains unclear, due to challenges in preserving the intact ligand in the final X-ray structure of the enzyme–DFMO complex. Consistently, only the hydrazide [22,23,25] or DFAcH [26] derivatives of the parent DFMO compound have been found in the available crystal structures. The resulting DFAcH species is tightly bound to the enzyme via a strong coordinate bond (N → Zn^2+^), in addition to other non-covalent bonds (NCIs) [25,26], and this interaction results in an essentially irreversible inhibition of HDAC6. Whether this peculiar inhibition mechanism has a role in determining the unique HDAC6 selectivity of these compounds is still unclear.

Notwithstanding the recent kinetic studies that consistently demonstrated that the DFMO moiety is first hydrolyzed within the active site [22,23,24,25,26], the reaction mechanisms proposed in the literature differ in whether the DFAcH intermediate dissociates from, or remains bound to, HDAC6 after the first hydrolysis step. In our previous publications [22,24], we argued that the second hydrolysis reaction requires the dissociation of the DFAcH intermediate and the entry of a second catalytic water molecule to restore the Zn^2+^ coordination sphere (mechanism 1 in Figure 1). In contrast, a detailed computational analysis carried out by Barinka and co-workers suggested that the DFMO and the DFAcH intermediate are directly hydrolyzed in situ, without requiring enzyme–intermediate complex disassembly [25]. Although this latter alternative reaction mechanism (mechanism 2 in Figure 1) is only supported by computational data, it appears plausible and may contribute, to some extent, to the reaction kinetics.

The first proposed mechanism [22,24] (Figure 1, mechanism 1) is supported by previous experimental data. In our reported kinetic studies, the DFAcH intermediate was isolated as a free species after spin column (size exclusion) chromatography [22,24], confirming its dissociation from the enzyme. Furthermore, König et al. reported that the incubation of fast-off oxadiazole inhibitors (e.g., trifluoromethyl oxadiazoles) with HDAC6 resulted in a faster formation of hydrazide compared to slow-off inhibitors [26]; this supports the idea that the DFAcH dissociation is the rate-determining step of the second hydrolysis. In addition, the kinetic study reported by Barinka and coworkers showed that during the reaction, the concentration of the intermediate species exceeded that of the enzyme, which indicates that some of it must exist in solution as a free species (see Figure 3B in [25]). Finally, in the same study, upon the near-complete depletion of DFMO, the conversion rate from DFAcH to hydrazide was stalled, which may probably reflect some competitive inhibition by the formed hydrazide [25]. All these results indicate the existence of the DFAcH intermediate as free species.

The abovementioned evidence clearly indicates the significant contribution of “mechanism 1” and that “mechanism 2” cannot be the sole mechanism taking place. However, the experimental and computational data collected so far do not definitively clarify which mechanism is predominant in the HDAC6-catalyzed hydrolysis of DFAcH. In this work, we provide additional nuclear magnetic resonance (NMR) spectroscopy and liquid chromatography–mass spectrometry (LC-MS) kinetic data, as well as the structure of a complex between zHDAC6-CD2 and a novel DFMO inhibitor, ITF7209 (Figure 2). Taken together, these data support an improved understanding of the reaction mechanisms of DFMO compounds.

## 2. Results and Discussion

### 2.1. Enzyme Kinetics on ITF5924 by NMR

In our previous publication, the conversion of a DFMO compound (ITF5924, Figure 2) to acylhydrazide and hydrazide by zHDAC6-CD2 was investigated using LC-MS analyses. Kinetic experiments were conducted using 1 µM zHDAC6-CD2 and 5 µM ITF5924. In those experiments we observed that the DFAcH concentration exceeded the zHDAC6-CD2 concentration, suggesting that the intermediate dissociated from the enzyme before the second hydrolysis step. To confirm these findings, we carried out a kinetic experiment using a higher DFMO-to-enzyme molar ratio.

For the first time, the reaction kinetics was monitored in situ using proton nuclear magnetic resonance (^1^H NMR) spectroscopy to quantitate the three species (DFMO, DFAcH, and hydrazide) without perturbing the chemical system, thus eliminating any potential variability and/or bias due to sample preparation. 

The kinetic experiment was conducted using 5 μM enzyme and 2 mM DFMO in deuterated assay buffer (25 mM d-Tris-DCl, 0.5 mM d-TCEP, pH 8.0). Quantitation of substrate, intermediate, and product in the mixture was achieved by integrating the singlets at 8.24, 8.20, and 8.18 ppm in the ^1^H NMR spectrum (Appendix A), which were assigned to the triazole protons of ITF5924, ITF6712, and ITF6715, respectively. The assignment was performed by separately analyzing the three species in the deuterated buffer.

The results of this experiment are represented in Figure 3, and they clearly show that the DFAcH intermediate is detected in solution at concentrations that largely exceed that of the enzyme (400 vs. 5 μM). Furthermore, the acylhydrazide-to-hydrazide molar ratio is higher at earlier time points. This evidence confirms that the DFAcH formed in the first hydrolysis step dissociates, at least partially, from zHDAC6-CD2 before it is transformed into the hydrazide product. Indeed, if mechanism 2 (Figure 1) was predominant, we would have observed a much lower concentration of free acylhydrazide, as the total concentration of this species would have been limited by the enzyme concentration.

Although conducted at higher concentrations and in a deuterated buffer, the NMR experiment qualitatively reproduced the results of the LC-MS experiments by Barinka and coworkers [25], where the acylhydrazide intermediate was detected at concentrations higher than zHDAC6-CD2 concentration (5 μM vs. 1 μM) with a comparable trend of the acylhydrazide-to-hydrazide molar ratio. We also note that, in this NMR experiment as well as in Barinka’s LC-MS experiment [25], upon the near-complete depletion of the DFMO, the rate of the second transformation is stalled, suggesting that, at higher concentrations, the hydrazide species effectively inhibits zHDAC6-CD2, thus preventing the quantitative conversion of DFAcH to hydrazide. As noted above, these results are not compatible with the hydrazide being formed by mechanism 2 only (Figure 1).

Accordingly, in agreement with our previous reports and Barinka’s experimental results, this NMR kinetic study provides additional evidence that mechanism 1 (Figure 1) significantly contributes to the formation of the hydrazide product.

### 2.2. Enzymatic Characterization of ITF7209 and Its Derivatives

ITF7209 (Figure 2), from our DFMO-focused medicinal chemistry program, was chosen as a new molecular model to further investigate the mechanism of action of this class of molecules. 

It should be noticed that this compound features a five-membered ring (thiophene) directly bound to the DFMO, a moiety that could induce different electronic effects on the ZBG and lead to a different binding geometry in the catalytic pocket, compared to the phenyl ring of ITF5924 (see section “X-Ray studies”). Moreover, the smaller cap-term of ITF7209 reduces the interaction with the binding site rim, which may minimize the potential impact of the ligand on the protein conformation.

ITF7209 is a potent inhibitor of both human full-length HDAC6 and zHDAC6-CD2 and is at least four orders of magnitude more potent on HDAC6 than HDAC1, taken as a representative of class I HDACs (Table 1). Analogously to that observed with ITF5924 and its metabolites [22], the DFAcH and hydrazide derivatives of ITF7209, ITF7738, and ITF7739, respectively, are much less potent than the parent compound (Table 1). Similar to all the other reported DFMOs [22,24,26,27], ITF7209 is confirmed to be a slow binding inhibitor of both HDAC6 enzymes (Appendix A). The non-linear relationship between calculated k_obs_ values and inhibitor concentration are consistent with the “induced fit” model (Appendix A) [28].

### 2.3. Enzyme Kinetics on ITF5924 and ITF7209 by LC-MS

The time-course assays describing the consumption of DFMO inhibitors by zHDAC6-CD2 reported in the literature were conducted under experimental conditions in which the ligand was present in a small molar excess over the enzyme, i.e., ten- [25] and five-fold [22]. Both studies revealed an exponential decay of DFMO, with the formation of the related DFAcH in the first few minutes, followed by the delayed appearance of the hydrazide final product. 

To better clarify the potential contribution of both plausible mechanisms to the reaction kinetics, the consumption time course of two DFMO compounds (ITF5924 and ITF7209) was evaluated under equimolar concentrations of DFMO and enzyme (5 µM), both in the presence and absence of a saturating concentration (100 µM) of a fast on/off hydroxamate competitor, ITF3756 (Appendix A). In order to achieve a fast and sufficient conversion of DFMO to DFAcH before the addition of the competitor, and to minimize its effect on the first hydrolysis step, ITF3756 was added after a very short DFMO–enzyme pre-incubation (1 min). Subsequently, at each time point (Figure 4), an aliquot of the reaction mixture was mixed with acetonitrile to unfold the protein and quench the enzymatic activity. Each aliquot was analyzed by liquid LC-MS to determine the compound concentrations.

The large excess of the competitor is expected to prevent the reassociation of the putatively dissociated DFAcH intermediate, thus “trapping” the free enzyme. Depending on the predominant mechanism, possible anticipated outcomes of these experiments in the presence of a strong competitor could be either a complete prevention of hydrazide formation (mechanism 1 in Figure 1), irrelevant effects on the second hydrolysis step (mechanism 2 in Figure 1), or a slowdown of the second nucleophilic attack (coexistence of both mechanisms).

In the case of ITF5924 (Figure 4A), the hydrolysis of DFMO in the absence of a competitor was very fast under these experimental conditions, and it substantially disappeared after 10 min, leading to the concomitant and proportional (1:1 ratio) increase in DFAcH (ITF6715). A relevant amount (13%) of hydrazide product (ITF6712) was detectable. Upon the near-complete consumption of the DFMO, the rate of both hydrolysis steps decreased, and after the complete disappearance of ITF5924 (t > 30 min), a slow conversion of ITF6715 to ITF6712 took place.

In an analogous experiment (Figure 4C), ITF7209 showed a slower rate of DFMO consumption and DFAcH (ITF7738) formation, and a delayed production of the hydrazide (ITF7739), when compared to ITF5924. 

Thus, ITF5924 and ITF7209 give rise to two distinct hydrolysis patterns. With ITF5924, the relatively large amount of the hydrazide product after 1 min may be indicative of a fast ring-opening followed by a rapid in situ nucleophilic attack (mechanism 2) [25]. The delayed decay of ITF6715 during the late stages may involve its dissociation, followed by the complex re-assembly where the catalytic conditions have been restored (mechanism 1). In contrast, the lack of hydrazide (ITF7739) at early times in the experiment with ITF7209 may suggest the predominance of mechanism 1.

The analogous time-course experiment of ITF5924 in the presence of a competitive inhibitor (Figure 4B) shows that the concentration of ITF5924 remains constant over time after the addition of the competitor. In contrast, the concentration of the DFAcH intermediate decreases, and that of the final hydrazide increases until the rate of the second transformation is stalled, probably due to the total conversion of bound intermediate to hydrazide. The very low affinity of the once-dissociated DFAcH intermediate (Table 1), coupled with the presence of an excess of the competitor, makes mechanism 2 more plausible; the formed DFAcH intermediate still present in the active site after the competitor addition is further hydrolyzed to hydrazide. This finding, in combination with previously reported data [22,25], suggests the potential coexistence of both reaction mechanisms for ITF5924 (mechanisms 1 and 2, Figure 1). 

On the other hand, the time-course experiment of ITF7209 in the presence of the competitor (Figure 4D) suggests that the relative contribution of the two mechanisms is different when compared to ITF5924. Interestingly, all species are substantially constant over time after the competitor addition, indicating that the mechanism involving the dissociation and rebinding of the intermediate DFAcH is predominant (mechanism 1) and that the in situ mechanism (mechanism 2) has a negligible impact on the final hydrolysis step. This interpretation is also consistent with the delayed formation of the DFAcH derivative of ITF7209 (Figure 4C), compared to that of the DFAcH derivative of ITF5924 (Figure 4A) in the absence of the competitor.

### 2.4. X-ray Studies

We hypothesized that the different behaviors of ITF5924 and ITF7209 observed in the LC-MS kinetic experiments may be caused by distinct conformations adopted by the two inhibitors in a complex with zHDAC6-CD2. Considering the structural information available in the literature about hydrazide/DFAcH-zHDAC6-CD2, it is noteworthy that no X-ray structure has ever been obtained for any unopened DFMOs to date.

A few recent studies [22,23,25,26] report the X-ray structure of the two hydrolyzed species of DFMO coprecipitated with zHDAC6-CD2, which confirm the structural features exhibited by hydroxamate-based ligand–enzyme complexes in the outer part of the catalytic pocket. The hydrophobic substructure (an aromatic ring or an aliphatic chain) lies in the crevice originated by Phe643 and Phe583 (zHDAC6-CD2 sequence numbering used hereon), which results in a relevant enthalpic contribution to the binding free energy (π-π interaction).

So far, the only available structural information on the binding conformer of DFAcH consisted of a crystal structure [26] and of the energetic profile obtained with an accurate computational investigation [25]. These calculations revealed that the rate-determining step is the DFMO ring opening, evolving into DFAcH. The resulting negative charge on N_4_ in the tetrahedral intermediate is stabilized by Zn-N_4_ bond formation (see atom numbering notation in Table 2 and Figure 5), allowing a large delocalization of the charge around the coordination sphere of the metal, whereas the residual proton from the catalytic water transferred to His573 (now cationic) provides electroneutrality. The recent release of the first complex (PDB code 8GD4) between zHDAC6-CD2 and a DFAcH [26] (compound 13 in Appendix A) broadly confirms this in silico prediction, showing a short Zn-N_4_ distance (2.06 Å, Table 2). The distance between His573 and the nitrogen atom bound to zinc (N_His573_⸳⸳⸳N_4_ = 3.62 Å) suggests a charge-assisted hydrogen bond (CAHB [29,30,31]), and an NCI with relevant strength, despite the long distance observed [32]. The distance between His574 and N_5_ (Table 2) characterizes a conventional hydrogen bond, where N_5_ acts as donor, in agreement with the predicted reaction energy profile. The carbonyl moiety derived [26] from the addition of the catalytic water (C_2_=O_3_) does not exhibit any interaction, whereas the fluorinated methyl group lands in a flat pocket delimited by Pro571, Cys584, and His574. The residue Tyr745 lies between a fluorine atom and the second carbonyl moiety of the ligand (C_6_=O_7_), suggesting its simultaneous engagement in a halogen bond [O_Tyr745_⸳⸳⸳F = 2.65 Å] [33] and a conventional hydrogen bond [34,35,36] with the carbonyl. 

In this work, we describe the X-ray crystal structure of the complex between zHDAC6-CD2 and the DFAcH of ITF7209 (ITF7738, Figure 2B, PDB Code 9EU0). Intriguingly, this DFAcH exhibits an unexpected binding geometry, with both nitrogen atoms (N_4_ and N_5_) lying far from dyadic histidine residues, despite a coordination sphere around the metal similar to the literature structure described above. However, the distance Zn-N_4_ is longer (2.32 Å) compared to the computed [25] (2.06 Å) and experimental [26] (2.02 Å, PDB 8GD4) complexes, suggesting a lower bond order. This feature is confirmed by the full analysis of bond distances in Table 2, where the C_2_-N_4_ bond in ITF7738 is shorter, close to amide average values, a detail that suggests a higher electron density localization in the ligand rather than along the N_4_-Zn(II) axis. Both carbonyl moieties of the ligand point toward the catalytic dyad His573-His574 [N_His573_⸳⸳⸳O_3_ = 2.50 Å; N_His574_⸳⸳⸳O_7_ = 2.37 Å]. The relatively short distances between heavy atoms suggest, especially for His574, a possible enhanced hydrogen bond (CAHB) if the original state of the dyad is not neutral, but positively charged. The relevant binding energy related to this interaction could compensate for the probable non-bonding effect connected to N_His573_⸳⸳⸳O_3_. In fact, the second proton derived from the catalytic water is presumably bound to N_5_, as indirectly indicated by the N_4_-N_5_ distance (1.40 Å), closer to a single rather than a double bond. It is worth noting that the hydrogen bond involves a positive histidine, and it can be regarded as an enhanced CAHB; moreover, the relatively reduced distance between heavy atoms suggests a relevant enthalpic contribution to binding, which is able to compensate the repulsive interaction involving His574. 

The CHF_2_ moiety lies in the flat pocket mainly originated by Pro571, which is buried in a local interaction network between the fluorine atoms and the closest residues. Lastly, Tyr745 exhibits a possible halogen bond (d = 2.37 Å) with the fluorinated methyl substructure and a hydrogen bond (d = 2.51 Å) with N_4_. 

The comparison between the new structure of ITF7738 and that of compound 13 [26] shows that the chemical nature of the “parent” DFMO does affect the binding mode of the resulting DFAcH. In turn, it is very possible that the specific conformation of the DFAcH-zHDAC6-CD2 complex dictates the relative prevalence of mechanism 1 or mechanism 2 (Figure 1).

## 3. Conclusions

In this work, we report a new complex between zHDAC6-CD2 and the ITF7209 derivative, which is a novel DFMO displaying a previously unseen binding mode. Additionally, we observed the kinetics of DFMO hydrolysis by NMR for the first time, complementing and strengthening our previously reported results [22]. 

Moreover, we discuss the two HDAC6-promoted DFMO hydrolysis reaction mechanisms present in the literature, which differ as to whether the DFAcH intermediate dissociates from HDAC6 after the first hydrolysis step (mechanism 1, Figure 1) or not (mechanism 2). Our goal was to verify the predominance of one mechanism over the other, or their potential coexistence. To this end, we designed a kinetic competition experiment to allow the direct observation of the potential DFAcH hydrolysis without its dissociation from the enzyme. The obtained results provided robust experimental evidence for the coexistence of the two alternative reaction mechanisms. Specifically, in the case of ITF5924, the reaction proceeds via both mechanisms, whereas the second hydrolysis of ITF7209 mostly proceeds via mechanism 1. 

In the context of this finding, there is an indication that the structure of the ligand may play a critical role in modulating the contribution of these two mechanisms to hydrazide product formation. The two DFMO molecules selected for the competition experiments (ITF5924 and ITF7209, Figure 2) differ in the presence of a phenyl or a thiophene ring, respectively, and in the size of their cap-term. Leveraging both current and prior crystallographic data [22,26], we hypothesize that larger orthosteric ligands more significantly disrupt the conformation of HDAC6 through interactions at the binding site rim. This perturbation may steer the ligand–protein complex towards different conformations, potentially favoring different hydrolysis pathways (mechanism 2 in Figure 1).

In conclusion, this research marks the first demonstration that the chemical nature and binding mode of a DFMO ligand to HDAC6 are critical determinants in the prevalence of one hydrolysis mechanism over the other. Our work contributes to a better understanding of the mechanism of HDAC6 inhibition by DFMOs, adding an important piece to unravel the intricacies of HDAC6oxadiazole interactions.

## 4. Materials and Methods

Full experimental procedures including synthesis of compounds, enzymatic measurements, NMR and LC-MS analysis, and X-ray crystallography are provided in the Appendix A.

## Figures and Tables

**Figure 1 ijms-25-05885-f001:**
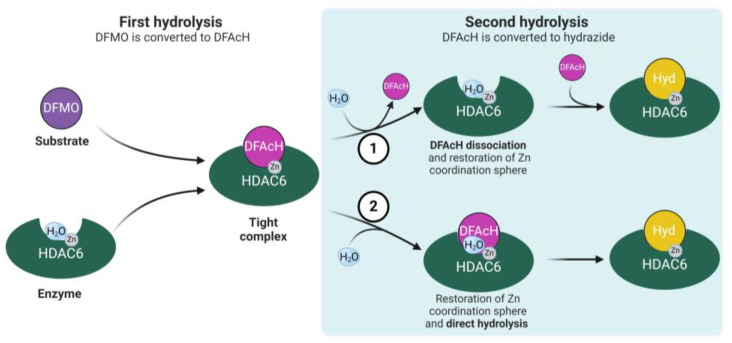
Diagram illustrating the potential reaction pathways of DFMO hydrolysis by HDAC6. The figure depicts the binding of DFMO to the HDAC6 active site, the first hydrolysis reaction, and the formation of the tight complex between the enzyme and DFAcH. Two alternative mechanisms for DFAcH hydrolysis to hydrazide are presented: (mechanism 1) DFAcH release, entry of a catalytic water molecule from the solvent to restore the tetrahedral zinc coordination sphere and intermediate rebinding for second hydrolysis reaction; (mechanism 2) entry of the second catalytic water molecule without intermediate dissociation and direct hydrolysis to hydrazide. Legend: difluoromethyl-1,3,4-oxadiazole (DFMO, purple), difluoroacetylhydrazide (DFAcH, light purple), hydrazide (Hyd, yellow), the zinc ion (Zn, grey), and water (H_2_O, light blue). Created with BioRender.com.

**Figure 2 ijms-25-05885-f002:**
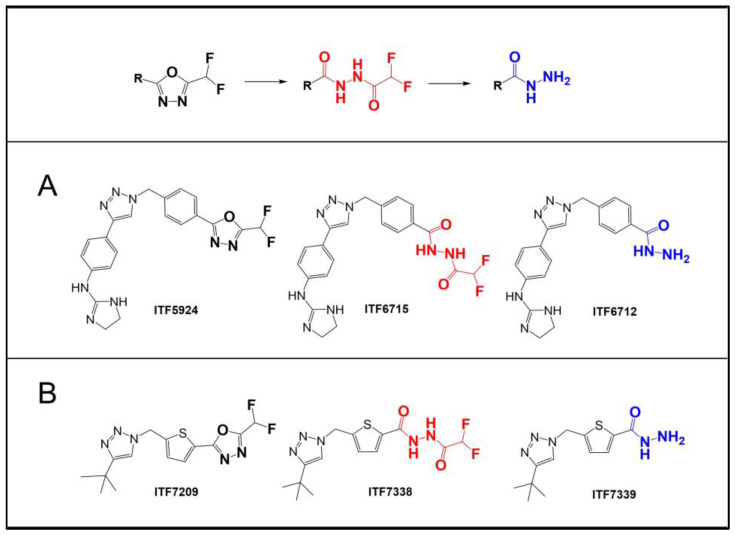
Structures of DFMO-selective HDAC6 inhibitors. (**A**) ITF5924, ITF6715, and ITF6712 are compounds 1, 2, and 3 from our previous publication [22]. (**B**) The DFMO-bearing compound ITF7209 and its acyl-hydrazide and hydrazide derivatives (ITF7738 and ITF7739), which are described for the first time in this work. The hydrazide group, the final metabolite of DFMO, is depicted in blue, while the intermediate difluoroacylhydrazide is in red.

**Figure 3 ijms-25-05885-f003:**
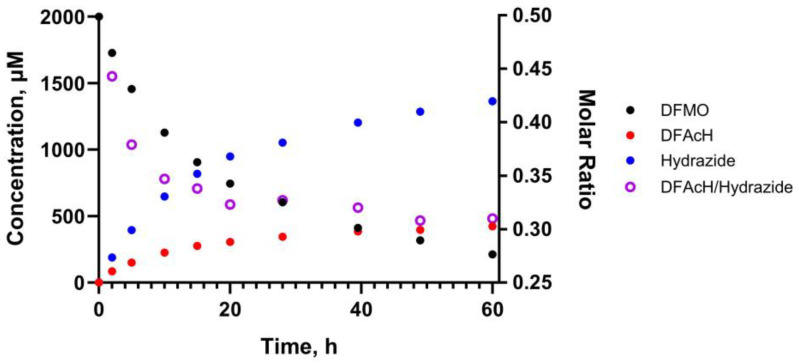
Hydrolysis of ITF5924 by zHDAC6-CD2 as detected by NMR. Evolution of DFMO (black), DFAcH (red), and hydrazide (blue) concentrations and the DFAcH-to-hydrazide molar ratio (purple open circles) over time (0 to 60 h). The kinetic experiment was conducted using 5 μM enzyme and 2 mM DFMO in deuterated assay buffer (25 mM d-Tris-DCl, 0.5 mM d-Tris(2-carboxyethyl)phosphine, pH 8.0). The experiment was carried out in singlicate (N = 1).

**Figure 4 ijms-25-05885-f004:**
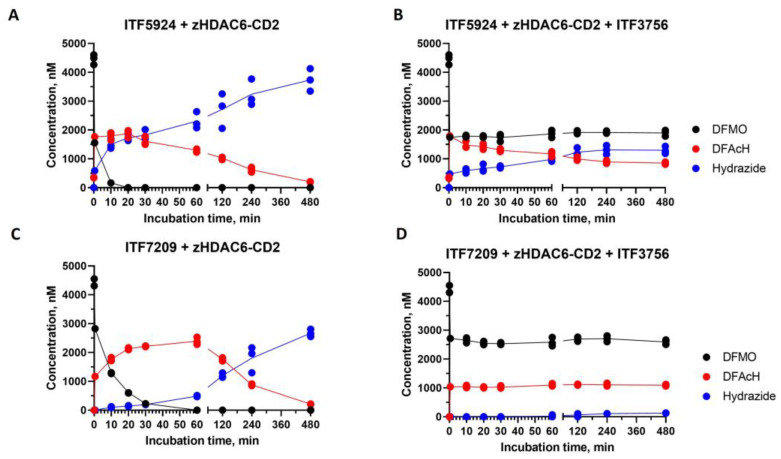
Hydrolysis of ITF5924 and ITF7209 by zHDAC6-CD2 as detected by LC-MS. The time course of 5 μM ITF5924 (**A**) or 5 μM ITF7209 (**C**) consumption during incubation at 25 °C with zHDAC6-CD2 (5 μM). Experiments like those shown in panel A and C were carried out in the presence of 100 μM ITF3756, which was added after 1 min of ITF5924-enzyme (**B**) or ITF7209-enzyme (**D**) pre-incubation. All the experiments were carried out in triplicate. The lines simply join the average values of the experimental points. The incomplete recovery could be attributed to the relatively high enzyme concentration, possibly resulting in coprecipitation phenomena upon the addition of acetonitrile.

**Figure 5 ijms-25-05885-f005:**
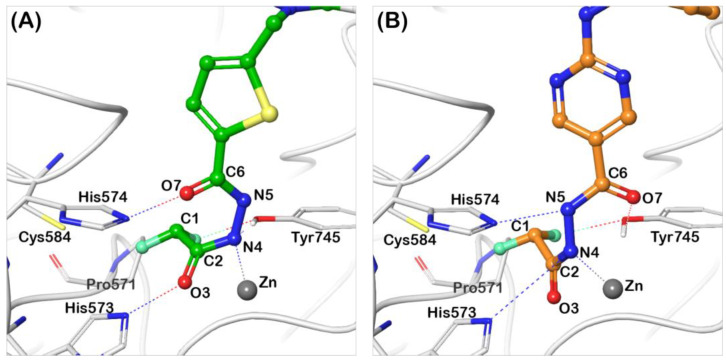
Structures of zHDAC6-CD2/DFAcH complexes. Comparison between the binding conformer of difluoroacetylhydrazide analogues of ITF7209 (panel (**A**), pdb code 9EU0) and compound 6 (panel (**B**), pdb code 8GD4 [26]).

**Table 1 ijms-25-05885-t001:** Inhibitory profile of ITF7209, ITF7738, and ITF7739 on HDACs.

HDACs IC_50_, nM			
Compounds	zHDAC6-CD2	HDAC6	HDAC1
**ITF7209**	6.7 ± 0.4	7.2 ± 0.6	N.A.
**ITF7738**	(72 ± 2) × 10^3^	N.A.	N.A.
**ITF7739**	(1.0 ± 0.1) × 10^3^	(5.1 ± 0.2) × 10^3^	N.A.
**ITF5924** ^a^	7.8 ± 0.4	7.7 ± 0.3	N.A.
**ITF6715** ^a^	(2.3 ± 0.1) × 10^3^	(1.6 ± 0.1) × 10^3^	N.A.
**ITF6712** ^a^	(9.7 ± 0.4) × 10	(3.3 ± 0.1) × 10^2^	N.A.

N.A.: not active (IC_50_ > 100 µM). ^a^ Data from [22] are included for comparison.

**Table 2 ijms-25-05885-t002:** Distance analysis of DFAcH moieties in experimental (8GD4 and 9EU0) and in silico (P_S1_) structures.

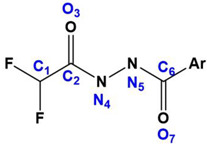				
Distances	Tabulated Values [Å] ^a^	8GD4 [Å]	9EU0 [Å]	P_S1_ [Å]
**C_2_=O_3_**	1.23	1.18	1.22	1.24
**C_2_-N_4_**	1.47 ^σ^/1.27 ^π^	1.42	1.35	1.39
**N_4_-N_5_**	1.47	1.43	1.40	1.42
**N_5_-C_6_**	1.47	1.44	1.35	1.34
**C_6_=O_7_**	1.23	1.20	1.22	1.27
**Zn(II)-N_4_**	-	2.02	2.32	2.06
**His573…N_4_**	-	3.62	-	3.01
**His574…N_5_**	-	2.78	-	2.68
**His573…O_3_**	-	3.01	2.50	-
**His574…O_7_**	-	-	2.37	-
**Tyr745…O_3_**	-	2.74	-	2.61
**Tyr745…N_4_**	-	3.75	2.51	4.25
**Tyr745…F**	-	2.65	2.37	3.41

^a^ Reference values of bond length from X-ray and neutron structures [37]. ^σ,π^ Refer to the tabulated lengths of single and double C-N bonds.

## Data Availability

All data are available in Appendix A.

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
