# Peer review of "Mechanistic and Structural Insights on Difluoromethyl-1,3,4-oxadiazole Inhibitors of HDAC6"

_ijms, 2024, doi:10.3390/ijms25115885_

Round 1
Reviewer 1 Report
Comments and Suggestions for Authors
In this manuscript, Cellupica and colleagues reported conprehensive characterization of the inhibition mechanism of HDAC6 by novel DFMO derivatives. The authors combined NMR, mass-spec and X-ray crystallography to comprehensively dissect and investigate the topic. I believe this paper is of very high quality and completely satisfies the publication standard on IJMS, thus I recommend this manuscript to be published in current form without revision.
I would only have two minor comments here:
- In the conclusion part, the authors will need to include more discussion about how the work presented would strengthen or prefer the proposed two enzymatic mechanisms of DFMO hydrolysis by HDAC6, which is not present in the current manuscript but was presented as one central question in the introduction.
- For the x-ray structures, he authors would perhaps want to improve their Rwork/Rfree values to around 0.2/0.26 since this would be the commonly accepted values for a ~1.8A resolution structure.
Reviewer 2 Report
Comments and Suggestions for Authors
The manuscript titled “Mechanistic and structural insights on difluoromethyl-1,3,4-oxadiazole inhibitors of HDAC6” by Cellupica, E.; et al. is a scientific work where the authors monitored the action of two different difluoromethyl-1,3,4oxadiazole molecules to inhibit the histone deacetylase-6 catalytical performance by a combination of complementary techniques. The most reventant finding obtained in this research could be interesting for a specialized target audience. Furthermore, the manuscript is generally well-written.
However, it exists some points that need to be addressed (please, see them below detailed point-by-point). The most relevant outcomes remarked by the authors can contribute in the growth of many fields by the better understanding of the underlying mechanisms involved in a panoply of human diseases where the histone regulation is directly connected. For this reason, I will recommend the present scientific manuscript for further publication in the International Journal of Molecular Sciences once all the below described suggestions will be properly fixed.
Here, there exists some points that must be covered in order to improve the scientific quality of the manuscript paper:
1) ABSTRACT. “Employing a combination of NMR and LC-MS kinetic (…)” (line 15). The full-name should be defined. Then, the abbreviations should be placed between brackets. This comment should be taken into account for the rest of the main manuscript body text.
2) INTRODUCTION. This section depicts the state-of-the-art of the examined field. “The mammalian histone deacetylase family (…) residues of histones, non-histone proteins and polyamines, thus playing a crucial role in various diseases” (lines 28-35). What diseases? Some information should provided in this regard. Then, it should be also mentioned the pivotal action of histones to form the protein machinery assemblies for DNA degradation [1] which could lead apoptotic diseases [2]..
[1] Novo, N.; Beyond a platform protein for the degradosome assembly: The Apoptosis-Inducing Factor as an efficient nuclease invovled in chromatinolysis. PNAS Nexus 2022, 2, pgac312. https://doi.org/10.1093/pnasnexus/pgac312.
[2] Wang, R.; et al. Targeting the DNA Damage Response for Cancer Therapy. Int. J. Mol. Sci. 2023, 24, 15907. https://doi.org/10.3390/ijms242115907.
3) MATERIALS & METHODS. The authors need to state the software tools used in this research to process the raw data.
4) RESULTS & DISCUSSION. Figure 3 (line 153). What is the population size (N) taken into account? The standard deviation (SD) bars should be added for each tested condition. Same comment for the Fig. 4 (line 262) and all the respective Figures that appear as Supplementary Information.
5) Then, did the authors observe any inhibition effect during the enzymatic kinetic assay? (due to the substrate, or any other type of competitive inhibitory phenomena).
6) Table 1 (line 200). Please, the authors should homogenize the significant figures.
7) CONCLUSIONS. Even if it is optional, it may be advisable to add a short statement to clearly state the most relevant outcomes found in this work. The authors should consider to add some potential future action lines to pursue this research.
Comments on the Quality of English LanguageThe manuscript is generally well-written albeit it may be desirable if the authors could recheck it in order to polish those final details susceptible to be improved.
